# Breast Reconstruction Use and Impact on Surgical and Oncologic Outcomes Amongst Inflammatory Breast Cancer Patients—A Systematic Review [†]

Ananya Gopika Nair [1], Gary Tsun Yin Ko [2], John Laurie Semple [1,3,4,5] and David Wai Lim [1,2,3,4,*]

1   Temerty Faculty of Medicine, University of Toronto, Toronto, ON M5S 1A8, Canada
2   Division of General Surgery, Department of Surgery, University of Toronto, Toronto, ON M5S 1A8, Canada
3   Women's College Research Institute, Women's College Hospital, Toronto, ON M5S 1A8, Canada
4   Department of Surgery, Women's College Hospital, Toronto, ON M5S 1A8, Canada
5   Division of Plastic, Reconstructive & Aesthetic Surgery, Department of Surgery, University of Toronto, Toronto, ON M5S 1A8, Canada
*   Correspondence: david.lim@wchospital.ca
†   This work was presented as an e-poster presentation at the Canadian Surgery Forum, 15–17 September 2022, in Toronto, Ontario, Canada and the American College of Surgeons Clinical Congress, 16–20 October 2022, in San Diego, California, USA.

**Abstract:** Breast reconstruction is generally discouraged in women with inflammatory breast cancer (IBC) due to concerns with recurrence and poor long-term survival. We aim to determine contemporary trends and predictors of breast reconstruction and its impact on oncologic outcomes among women with IBC. A systematic literature review for all studies published up to 15 September 2022 was conducted via MEDLINE, Embase, and the Cochrane Library. Studies comparing women diagnosed with IBC undergoing a mastectomy with or without breast reconstruction were evaluated. The initial search yielded 225 studies, of which nine retrospective cohort studies, reporting 2781 cases of breast reconstruction in 29,058 women with IBC, were included. In the past two decades, immediate reconstruction rates have doubled. Younger age, higher income (>USD 25,000), private insurance, metropolitan residence, and bilateral mastectomy were associated with immediate reconstruction. No significant difference was found in overall survival, breast cancer-specific survival or recurrence rates between women undergoing versus not undergoing (immediate or delayed) reconstruction. There is a paucity of data on delayed breast reconstruction following IBC. Immediate breast reconstruction may be a consideration for select patients with IBC, although prospective data is needed to clarify its safety.

**Keywords:** breast reconstruction; inflammatory breast cancer; survival; immediate reconstruction

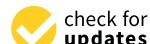



## 1. Introduction

Inflammatory breast cancer (IBC) is a rare presentation that represents only 1–5% of all breast cancer diagnoses, with a disproportionately high mortality rate of 8–10% [1,2]. The clinical presentation of IBC is unique, characterized by erythema and edema developing rapidly and involving at least one-third of the breast ('peau d'orange' appearance) [3]. Survival after IBC has been poor, with five-year survival at less than 10% and high local recurrence rates at 64.8% [4,5]. Given these concerns, along with issues related to surgical margin positivity and potential delay of adjuvant treatments from surgical complications, immediate breast reconstruction has been considered a relative contraindication in IBC patients [6,7].

With advances in treatment for IBC, particularly the recommended trimodality therapy of neoadjuvant systemic chemotherapy followed by modified radical mastectomy and radiation therapy, five-year survival rate, distant metastasis-free survival rate, and

locoregional control have improved to 30–70%, 47%, and 84%, respectively [2,8–10]. In a subset of IBC patients with a pathologic complete response to neoadjuvant chemotherapy, the five-year local control rate was as high as 95% [11]. Notably, recent studies on IBC have found no association between having immediate breast reconstruction and worse oncologic outcomes [12–14].

Current recommendations propose delaying reconstruction until after definitive treatment [15]. Nevertheless, an increasing proportion of women are choosing to undergo breast reconstruction at the time of initial surgical therapy [2,13,14]. Taking into consideration the improving survival of women with IBC and the potential psychosocial benefits of breast reconstruction, the role of immediate breast reconstruction deserves further study. We evaluated the literature to determine contemporary trends and predictors of breast reconstruction use among women with IBC and determined the impact of breast reconstruction on their survival.

## 2. Methods

### 2.1. Study Strategy

The protocol for this systematic review was registered with PROSPERO (International Prospective Register of Systematic Reviews), Protocol CRD42023400859. Our systematic review was performed according to Preferred Reporting Items for Systematic Reviews and Meta-Analyses (PRISMA) guidelines.

A systematic search of MEDLINE, EMBASE, PubMed, and the Cochrane Central Register of Controlled Trials, from inception up to and including 15 September 2022, was performed. To ensure a wide array of articles was identified, we used a comprehensive search strategy incorporating MeSH headings, keywords, and free text relating to inflammatory breast cancer, reconstruction surgery, and cancer outcomes (Table S1). The reference lists of included studies and relevant articles were also searched to identify any eligible publications.

### 2.2. Eligibility Criteria

Studies were included based on the following criteria: primary research on adult patients diagnosed with IBC undergoing a mastectomy, with or without breast reconstruction. Studies were excluded based on the following criteria: review articles or lacking original data, conference reports or published abstracts without accompanying complete articles, articles reporting on lumpectomy, articles reporting only on surgical technique, studies using non-human participants or cadavers, and articles in non-English languages. No restrictions were placed on the type of mastectomy or reconstruction surgery. Patients were not excluded based on the completion of their neoadjuvant chemotherapy, adjuvant radiation therapy, or their reconstruction timeline (delayed versus immediate).

### 2.3. Study Selection

After removing duplicates from the initial search, titles and abstracts were manually screened by two reviewers (AGN, GTYK) using an explicit pre-determined criterion. Where eligibility remained unclear, the articles were assessed by an additional reviewer (DWL) with a final decision being reached by consensus.

### 2.4. Data Extraction

Data were extracted from each eligible study by one reviewer (AGN) using a standardized electronic data collection form. Information about the study, including author, year of publication, sample size, and patient demographics, was collected. Additional information on preoperative factors and patient outcomes relating to breast reconstruction were also recorded. Only published data were used.

*2.5. Definitions of Outcomes of Interest*

The primary indication for mastectomy was defined as those performed therapeutically for any non-metastatic IBC. Those with a prophylactic indication were mastectomies performed for aesthetic symmetry and/or future risk reduction of breast cancer, owing to recognized genetic abnormalities, and significant family history. Pre-operative factors were examined to understand contemporary trends and evaluate predictors associated with contralateral prophylactic mastectomy and breast reconstruction after mastectomy. These included demographic (age, year of diagnosis, race, ethnicity), clinicopathologic (comorbidity status, pathologic grade, stage, burden of nodal disease, receptor status) and socioeconomic factors (median income, level of education, insurance, metropolitan living). Surgical outcomes were evaluated based on post-operative complications (e.g., wound infection, healing issues, debridement), hospital readmission, length of hospital stay and mortality within 30 days of breast reconstruction. Oncologic outcomes were evaluated by overall survival, overall mortality, breast cancer-specific survival, time to adjuvant radiotherapy, and locoregional or distant recurrence.

*2.6. Quality Assessment*

Risk-of-bias assessments for observational or non-randomized surgical studies were performed using the adapted Newcastle–Ottawa quality assessment scale. The assessment was based upon selection (representativeness/selection of the cohorts, demonstration of prospective design), comparability (statistical adjustment confounders), and outcome (outcome ascertainment, sufficiency of follow-up) domains [16]. The Newcastle-Ottawa scale domains were applied to each included study by the reviewer (AGN) responsible for collecting and extracting data from the articles, and the senior author (DWL). No study was excluded based on quality assessment.

*2.7. Data Synthesis and Analysis*

The primary analysis evaluates predictors for breast reconstruction and compares the surgical and oncologic outcomes between patients with and without breast reconstruction. A narrative synthesis was used to describe all comparisons between outcomes of interest. Odds ratios (ORs), hazard ratios (HR), and 95% confidence intervals (95% CIs) were preferentially extracted from the data with *p* values < 0.05 considered statistically significant. A quantitative meta-analysis was not conducted, given the heterogeneity between included studies. A formal assessment of publication bias was not performed, as fewer than ten studies were included.

## 3. Results

The initial electronic search, removing duplicates, identified 186 potentially eligible studies. After screening titles and abstracts, 167 studies were excluded for not fulfilling eligibility criteria. Of the 19 full texts retrieved and evaluated, ten were excluded (Figure 1). A total of nine studies were eligible for final inclusion [13,14,17–23]. Characteristics of the included studies are summarized in Table 1. All but one study was published in the last decade, many using patient data from 1987 to 2016. The studies were either single-institution retrospective studies (*n* = 4) or used population-based data from national registries including the National Cancer Database (NCDB, *n* = 2) and Surveillance, Epidemiology, and End Results (SEER, *n* = 3) database. All the studies were conducted in the United States, except for two which were completed by authors in Canada and China but using data from the United States.

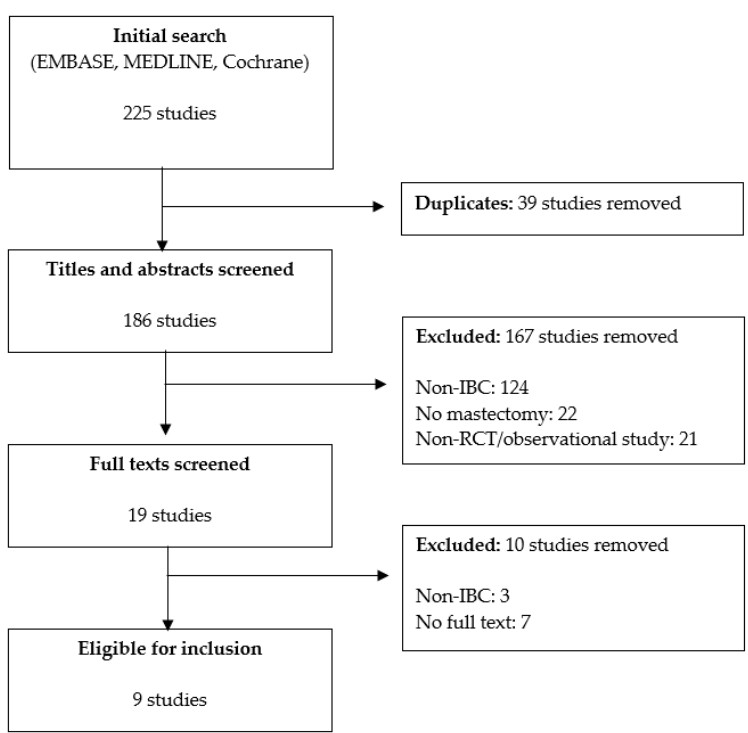

**Figure 1.** PRISMA flow diagram.

**Table 1.** Patient demographics of included studies.

| Study Author, (Publication Year), Article Country of Origin | Methodology (Years Analyzed) | Sample Size | Median Age of Diagnosis, Years (Range) | Race | Insurance Status | Median Income (USD) | Patient Location |
|---|---|---|---|---|---|---|---|
| Chang et al. [17], (2015) USA | Retrospective (2000–2012) | 830 | 48.0 (27–65) | NR | NR | NR | NR |
| Chen et al. [14], (2017) China | Retrospective SEER (1998–2013) | 3224 | 53.0 (22–90) | White: 2723 Black: 434 Asian/Indian: 209 Unk: 8 | NR | NR | NR |
| Chin et al. [18], (2000) USA | Retrospective (1987–1997) | 23 | 49 (35–62) | NR | NR | NR | NR |
| Hoffman et al. [19], (2021) USA | Retrospective NCDB (2004–2016) | 6589 | 54.9 ± 12.1 (mean ± SD) | White: 5358 Black: 960 Asian: 152 Other/Unk: 119 | Uninsured: 304 Private: 3827 Medicaid: 836 Medicare: 1444 Other govt: 85 Unk: 93 | <40,227: 1250 40,227–50,353: 1469 50,354–63,332: 1542 >63,333: 2217 Other/Unk: 111 | Metro: 5356 Urban: 948 Rural: 136 Other/Unk: 149 |
| Karadsheh et al. [13], (2021) USA | Retrospective NCDB (2004–2016) | 12,544 | 56.9 mean | NR | Uninsured: 521 Private: 6665 Medicaid: 1569 Medicare: 3484 Other govt: 143 Unk: 162 | <38,000: 2309 38,000–47,999: 2989 48,000–62,999: 3422 >63,000: 3748 Unk: 76 | Large metro: 6382 Metro: 3810 Urban: 746 Rural: 1293 Unk: 313 |
| Nair et al. [23], (2022) Canada | Retrospective SEER (2004–2015) | 4076 | 55.1 ± 12.5 (mean ± SD) | White: 3242 Black: 569 Southeast Asian: 112 East Asian: 58 Other/Unk: 95 | Uninsured: 304 Private: 3827 Medicaid: 836 Medicare: 1444 Other govt: 85 Unk: 93 | <55,000: 1032 55,000–74,999: 1993 >75,000: 1050 Unk: 1 | Metro: 3209 Non-metro: 856 Unk: 11 |
| Nakhlis et al. [20], (2019) USA | Retrospective (1997–2016) | 240 | 51.0 (28–78) | NR | NR | NR | NR |
| Patel et al. [21], (2018) USA | Retrospective SEER-Medicare (1991–2009) | 1472 | 75.5 (65–103) | White: 1235 Black: 153 Other: 84 | Medicare: 1472 | <25,000: 155 25,000–50,000: 844 50,000–75,000: 348 >75,000: 125 | Large metro: 750 Metro: 480 Urban: 84 Rural: 158 |
| Simpson et al. [22], (2016) USA | Retrospective (2006–2014) | 60 | 55 (33–67) | NR | NR | NR | NR |

Abbreviations: govt: government; NCDB: National Cancer Database; NR: not reported; Medicare: federal health insurance for individuals >65 years old; Medicaid: federal and state program providing health coverage to individuals of lower economic status; Metro: metropolitan; SD: standard deviation; SEER: Surveillance, Epidemiology, and End Results; Unk: unknown.

### 3.1. Methodological Quality

All studies eligible for inclusion were non-comparative retrospective studies, allowing for a minimum score of 0 and a maximum score of 9 using the Newcastle–Ottawa scale. Overall, the quality of reporting amongst the studies was good, with a median score of eight (range 5–8, Table 2). Representativeness and selection of the cohorts, ascertainment of patient details, assessment of outcomes, and length and adequacy of follow-up were generally well reported. Three single-institution studies were deemed poor quality simply because they were not able to control for age or other confounding variables such as income. The retrospective nature of all the studies and the inconsistent control for or comment upon potential confounding variables impacted the overall quality assessment of the studies. Notably, SEER and NCDB do not capture delayed reconstruction. Information on delayed reconstruction specific to IBC are only provided from the single-institution retrospective studies.

**Table 2.** Adapted Newcastle–Ottawa quality assessment scale for non-randomized cohort studies.

| Study | Selection | | | | Comparability | | Outcome | | | Score | Overall Quality |
| | Representative Exposed Cohort | Selection of Non-Exposed Cohort | Ascertainment of Exposure | Prospective | Control for Age, Stage | Other Controls | Outcome Assessment | Length Follow-Up | Adequate Follow-Up | | |
|---|---|---|---|---|---|---|---|---|---|---|---|
| Chang et al., 2015 [17] | (A) * | (A) * | (A) * | (B) | (C) | (C) | (B) * | (A) * | (A) * | 6 | Poor |
| Chen et al., 2017 [14] | (A) * | (A) * | (A) * | (B) | (A) * | (B) * | (B) * | (A) * | (A) * | 8 | Good |
| Chin et al., 2000 [18] | (A) * | (C) | (A) * | (B) | (C) | (C) | (B) * | (A) * | (A) * | 5 | Poor |
| Hoffman et al., 2021 [19] | (A) * | (A) * | (A) * | (B) | (A) * | (B) * | (B) * | (A) * | (A) * | 8 | Good |
| Karadsheh et al., 2021 [13] | (A) * | (A) * | (A) * | (B) | (A) * | (B) * | (B) * | (A) * | (A) * | 8 | Good |
| Nair et al., 2022 [23] | (A) * | (A) * | (A) * | (B) | (A) * | (B) * | (B) * | (A) * | (A) * | 8 | Good |
| Nakhlis et al., 2020 [20] | (A) * | (A) * | (A) * | (B) | (C) | (C) | (A) * | (A) * | (A) * | 6 | Poor |
| Patel et al., 2018 [21] | (A) * | (A) * | (A) * | (B) | (A) * | (B) * | (A) * | (A) * | (A) * | 8 | Good |
| Simpson et al., 2016 [22] | (A) * | (A) * | (A) * | (B) | (C) | (B) * | (A) * | (A) * | (A) * | 7 | Good |

* Point awarded. Selection: (1) Representativeness of the exposed cohort: (A) * truly representative of the average patient with comorbidity in the community, (B) * somewhat representative of the average patient with comorbidity in the community, (C) selected group of users, (D) no description. (2) Selection of the non-exposed cohort: (A) * drawn from the same community as the exposed cohort, (B) drawn from a different source, (C) no description. (3) Ascertainment of exposure: (A) * secure record, (B) * structured interview, (C) written self-report, (D) no description. (4) Demonstration that the outcome of interest was not present at the start of the study: (A) * yes (prospective), (B) no (retrospective). Comparability: (1) (A) * study controls for age or stage (or tumor size and lymph node status), (B) * study controls for other confounding factors (preoperative treatment status, financial means), (C) study does not control for confounding variables. Outcome: (1) Assessment of outcome: (A) * independent blind assessment, (B) * record linkage, (C) self-report, (D) no description. (2) Follow-up long enough for outcomes to occur: (A) * yes (>48 months from completion of treatment), (B) no, (C) no statement. (3) Adequacy of follow-up of cohorts: (A) * complete follow-up with all subjects accounted for, (B) * small number lost (<20%) and description provided of those lost, (C) follow-up rate < 80% and no description of those lost, (D) no statement. Overall quality: Good: 3 or 4 stars in selection domain AND 1 or 2 stars in comparability domain AND 2 or 3 stars in outcome domain. Fair: 2 stars in selection domain AND 1 or 2 stars in comparability domain AND 2 or 3 stars in outcome domain. Poor: 0 or 1 star in selection domain OR 0 stars in comparability domain OR 0 or 1 stars in outcome domain.

### 3.2. Patient Population

The number of patients in the studies ranged from 23 to 12,544 patients. The median age was 54.9 years, ranging from 28–103 years. A summary of patient characteristics is presented in Table 1. Among the included women, 2781 (9.6%) underwent breast recon-

struction, with 1117 (40.2%) immediate and 93 (3.3%) delayed reconstruction cases. Most women (*n* = 969, 34.8%) had autologous reconstruction, 638 (22.9%) women underwent alloplastic reconstruction, and 237 (8.5%) women received a combination of both (e.g., latissimus dorsi flap with tissue expander) (Table 3). The timeline for breast reconstruction (immediate versus delayed) for 1571 women (56.5%) was unspecified. Most women were documented as receiving neoadjuvant chemotherapy (*n* = 23,479; 80.8%) and adjuvant radiation therapy (*n* = 23,101; 79.5%), respectively; 21,246 (73.1%) patients underwent a unilateral mastectomy, while 6186 (21.3%) also had a contralateral prophylactic mastectomy (Table 3).

**Table 3.** Patient inflammatory breast cancer treatment characteristics.

| Study | BRC Surgery | BRC Type | Treatment History |
|---|---|---|---|
| Chang et al., (2015) [17] | NBRC: 771 (92.9%) IBRC: 7 (0.8%) DBRC: 52 (6.3%) | 59 ATL (100.0%) | 59 CTX, XRT (100%) 45 SMX (5.4%) 12 SMX + CPM (1.4%) 1 DMX (for bilateral breast cancer) (0.1%) 1 SMX + CL (0.1%) |
| Chen et al., (2017) [14] | NBRC: 2960 (91.8%) IBRC: 264 (8.2%) | 112 ATL (42.4%) 68 ALP (25.8%) 30 CMB (11.4%) 54 unspecified (20.4%) | 3224 CTX, XRT (100%) 2632 SMX (81.6%) 592 SMX + CPM (18.4%) |
| Chin et al., (2000) [18] | IBRC: 14 (60.9%) DBRC: 9 (39.1%) | 20 ATL (87.0%) 3 ALP (13.0%) | 22 CTX (95.7%) 14 XRT (60.9%) 23 SMX (100%) |
| Hoffman et al., (2021) [19] | NBRC: 5954 (90.4%) IBRC: 635 (9.6%) | 250 ATL (39.4%) 171 ALP (26.9%) 64 CMB (10.1%) 150 unspecified (23.6%) | 6589 CTX, XRT (100%) 4031 SMX (61.2%) 1705 SMX + CPM (25.9%) 853 unknown if CPM performed (12.9%) |
| Karadsheh et al., (2021) [13] | NBRC: 11 237 (89.6%) IBRC:1307 (10.4%) | 491 ATL (37.6%) 374 ALP (28.6%) 142 CMB (10.9%) 300 unspecified (22.9%) | 12,252 CTX, XRT (97.7%) 9579 SMX (76.4%) 2965 SMX + CPM (23.6%) |
| Nair et al., (2022) [23] | NBRC: 3688 (90.5%) IBRC: 388 (9.5%) | NR | 3181 SMX (78.0%) 895 SMX + CPM (22.0%) |
| Nakhlis et al., (2019) [20] | NBRC: 200 (83.3%) IBRC: 13 (5.4%) DBRC: 27 (11.3%) | 29 ATL (72.5%) 9 ALP (22.5%) 1 CMB (2.5%) 1 unspecified (2.5%) | 240 ptx CTX (100%) 240 SMX (100%) |
| Patel et al., (2018) [21] | NBRC: 1428 (97.0%) IBRC: 44 (3.0%) | NR | 957 CTX (65.0%) 893 XRT (60.7%) 1472 SMX (100%) |
| Simpson et al., (2016) [22] | NBRC: 39 (65.0%) IBRC: 16 (26.7%) DBRC: 5 (8.3%) | 8 ATL (38.1%) 13 ALP (61.9%) | 60 CTX, XRT (100%) 43 SMX (71.7%) 17 SMX + CPM (28.3%) |

Abbreviations: ALP: alloplastic; ATL: autologous; CL: contralateral lumpectomy; CMB: combined alloplastic and alloplastic (e.g., latissimus dorsi flap with tissue expander); CPM: contralateral prophylactic mastectomy; CTX: chemotherapy; DBRC: delayed breast reconstruction; DMX: double mastectomy; IBRC: immediate breast reconstruction; NBRC: no breast reconstruction; NR: not reported; ptx: patients; SMX: single ipsilateral mastectomy; XRT: radiation therapy.

*3.3. Contemporary Trends in Breast Reconstruction Use and Contralateral Prophylactic Mastectomy*

Four studies [13,14,19,23] evaluating breast reconstruction trends between 1998 and 2016 found that reconstruction rates increased from 6.2% to 15.3% (Table 4). In parallel, over the last two decades, contralateral prophylactic mastectomy (CPM) rates have increased from 6.1% to 29.6% (Table 4) [13,14,23].

**Table 4.** Patterns and factors associated with having breast reconstruction.

| Study | BRC Rate | CPM Rate | Demographic Factors | Tumor/Pathology Factors | Socioeconomic Factors |
|---|---|---|---|---|---|
| Chen et al., (2017) [14] | 6.5% to 10.9% ($p = 0.011$) | 6.1% to 29.4% ($p < 0.001$) | NR | DMX or SMX + CPM increased BRC ($p < 0.001$) | NR |
| Hoffman et al., (2021) [19] | 61% increase from 6.3% to 10.1% ($p < 0.001$) | NR | Younger age—BRC (49.5 ± 10.4) NBRC (55.4 ± 12.1) [OR 1.83, 95% CI 1.39–2.41] $p < 0.01$ No differences in race or comorbidity status | Bilateral mastectomy [OR 1.67, 95% CI 1.28–2.17] $p < 0.01$ No differences in tumor characteristics, immunohistochemical subtype, clinical, pathologic stage, tumor grade, margin status, lymphovascular invasion. | Private insurance—BRC (74.3%) vs. NBRC (56.3%) [OR 2.49, 95% CI 1.65–3.77]; $p < 0.01$ Not on Medicare—BRC (9.6%) vs. NBRC (23.2%); $p < 0.01$ Higher median income (>USD 63,333)—BRC (49.0%) vs. NBRC (32.0%); $p < 0.01$ Metropolitan region—BRC (90.4%) vs. NBRC (80.3%) [OR 2.34, 95% CI 1.36–4.03]; $p < 0.01$ Higher education; $p < 0.01$ |
| Karadsheh et al., (2021) [13] | 7.3% to 12.3% ($p < 0.001$) | 11.7% to 26.3% ($p < 0.001$) | Younger age—BRC (50.8 yrs) vs. NBRC (57.2 yrs); $p < 0.001$ Lower CCI; $p < 0.001$ | Lower pathologic stage; $p < 0.001$ Lower pathologic nodal status; $p < 0.001$ CPM; $p < 0.001$ | Private insurance—BRC (71.8%) vs. NBRC (51.0%); $p < 0.001$ Higher median income (>USD 63,000)—BRC (44.1%) vs. NBRC (28.2%); $p < 0.001$ Metropolitan region—BRC (65.3%) vs. NBRC (49.2%); $p < 0.001$ |
| Nair et al., (2022) [23] | 6.2% to 15.3% ($p < 0.001$) | 12.9% to 29.6% ($p < 0.001$) | Younger age ($p < 0.0001$) No difference in race or ethnicity | CPM, $p < 0.0001$ No differences in tumor size, grade, nodal burden, or receptor status. | High income (>USD 75,000); $p = 0.027$ Metropolitan living; $p < 0.02$ |
| Patel et al., (2018) [21] | NR | NR | Younger age—BRC (72.6 yrs) vs. NBRC (75.6 yrs); $p = 0.008$ Lower CCI; $p = 0.29$ | NR | Median income (USD 25,000–75,000) ($p = 0.024$) In MV analysis, income was an independent predictor ($p = 0.047$) |

Abbreviations: BRC: breast reconstruction; CI: confidence interval; CCI: Charlson Comorbidity Index; CPM: contralateral prophylactic mastectomy; DMX: double mastectomy; SMX: single ipsilateral mastectomy; NBRC: no breast reconstruction; NR: not reported; MV: multivariate; OR: odds ratio. Chin et al., (2000) [18], Chang et al., (2015) [17], Simpson et al., (2016) [22], and Nakhlis et al., (2019) [20] did not report on predictive factors for breast reconstruction.

### 3.4. Factors Associated with Breast Reconstruction Use

The five population-based studies evaluated preoperative factors predicting having breast reconstruction use [13,14,19,21,23]. Younger patients and those with higher median income (USD 25,000–75,000), access to private insurance, not being on Medicare, and completion of higher education were more likely to undergo breast reconstruction. Some studies identified bilateral mastectomy (single mastectomy with a contralateral prophylactic mastectomy) to be strongly associated with breast reconstruction [13,14,19,23]. Three studies found living in a metropolitan setting was associated with breast reconstruction compared with living in a rural setting [13,19,23]. Other demographic and socioeconomic factors of race, ethnicity and marital status were not associated with breast reconstruction (Table 4).

The data on clinicopathological predictors of reconstruction use is mixed. Comorbidity status and tumor characteristics (lower pathologic stage, lower burden of nodal disease) were found to increase the likelihood of breast reconstruction in one study [13], while three studies found no such association [19,21,23] (Table 4).

### 3.5. Breast Reconstruction and Oncologic Outcomes

Post-Operative Complications

Five studies (four single institution, one population-based) reported on post-surgical complications among patients with and without breast reconstruction [17,19–22] (Table 5). In the study by Chang et al., 21 out of 59 patients experienced a complication after reconstruction [17]. The complications include delayed wound healing, hernia, flap loss and other medical issues [17]. Among 61 patients who had breast reconstruction in studies by Nakhlis et al. and Simpson et al., 14 (23.0%) experienced complications such as revision surgery ($n = 4$), delayed wound healing ($n = 3$), tissue loss ($n = 2$), expander removal ($n = 2$) and other medical issues ($n = 3$) (Table 5) [20,22]. In Patel et al., three of 44 patients (7%) experienced an unspecified mechanical complication due to the breast implant. Length of stay was longer among immediate breast reconstruction patients than those having no reconstruction (2.4 vs. 1.4 days; $p < 0.01$) in the study by Hoffman et al. However, there was no difference in 30-day re-admission rates ($p = 0.94$) and 30-day or 90-day mortality ($p = 0.12$) [19].

**Table 5.** Post-mastectomy patient complications and time to adjuvant therapy.

| Study | Post-Operative Complication | Time to Adjuvant XRT |
|---|---|---|
| Chang et al., (2015) [17] | 21 complications out of 59 patients (7 delayed flap healing, 5 abdominal donor site healing, 1 abdominal donor site hernia, 1 total flap loss, 4 fat necrosis, 4 pedicle thrombosis, 2 medical complications) | NR |
| Hoffman et al., (2021) [19] | BRC longer length of stay than NBRC 2.4 ($\pm$8.0) days vs. 1.4 ($\pm$3.8) days; $p < 0.01$ No difference in 30-day readmission rates; $p = 0.94$ No difference in 30-day or 90-day mortality; $p = 0.12$ | BRC: 8 weeks (6–10 weeks) NBRC: 7 weeks (5–10 weeks) No difference in time to XRT between BRC and NBRC group $p = 0.93$ |
| Nakhlis et al., (2019) [20] | 1/13 IBRC (TRAM flap necrosis) 7/27 DBRC (1 abdominal donor site dehiscence, 1 chronic hematoma, 1 infection requiring hospitalization, 1 donor site hernia, 3 fat necrosis/capsular contracture) | NBRC: 3 months (1–10 mos) IBRC: 56.5 days (23–123 days) DBRC: 3 months (2–10 mos) |
| Patel et al., (2018) [21] | BRC: 3/44 implant-related complications (7%, $p < 0.003$ versus 0/1428 NBRC)) | NR |
| Simpson et al., (2016) [22] | NBRC: 1/39 (2.6%)—hematoma IBRC: 6/16 (37.5%)—2 infection, 1 expander removal, 2 tissue loss DBRC: 0/5 (0.0%) $p = 0.006$ | NBRC: 42 days IBRC: 52.5 days DBRC: 45 days BRC not associated with delay to XRT $p = 0.86$ |

Abbreviations: BRC: breast reconstruction; CPM: contralateral prophylactic mastectomy; DBRC: delayed breast reconstruction; IBRC: immediate breast reconstruction; mos: months; NBRC: no breast reconstruction; NR: not reported; SBRC: single breast reconstruction; XRT: radiation therapy; yrs: years. Chin et al. [18], (2000), Chen et al., (2017) [14], Karadsheh et al., (2021) [13], and Nair et al., (2022) [23] did not report on post-operative complications or time to adjuvant radiotherapy.

### 3.6. Time to Adjuvant Therapy

Three studies reported the impact of breast reconstruction on time to adjuvant radiation therapy [19,20,22] (Table 5). Breast reconstruction, particularly immediate breast reconstruction, increased the risk of postoperative complications (infections, delayed wound healing, longer hospital stays) compared with no reconstruction [19,20,22]. However, there was no association between breast reconstruction and delays in starting adjuvant treatment [19,22]. Time to post-mastectomy radiation was between 42–56 days for women not having breast reconstruction versus 52.5–56.5 days for women having immediate breast reconstruction [19,20,22].

### 3.7. Overall Survival and Mortality

All studies evaluated the relationship between breast reconstruction and overall survival (OS). The median follow-up and median survival ranged from 27.6–68.4 months [14,17–19,22,23] and 22.0–87.0 months, respectively [13,17,20]. None of the included studies found breast reconstruction to negatively impact OS; three studies found it to improve OS while others found no association between breast reconstruction and OS (Table 6).

**Table 6.** Reconstruction and oncologic outcomes.

| Study | Median Follow-Up (Months) | OS | OM | BCSS | Recurrence |
|---|---|---|---|---|---|
| Chang et al., (2015) [17] | 43.9 (5.1–140) | Median survival: 44.0 mos (38.6–48.6 mos) | BRC: 13.6% (8/59 deaths— 7 IBC related, 1 other cause) NBRC: NR | 11.9% (7/59 patients) | 1 LRR at 7 mos |
| Chen et al., (2017) [14] | 47.0 (4–203) | 5-year OS: 55.9% 3-year OS improved over time (62.8% to 78.5%; $p < 0.0001$) | NR | 5-year BCSS 59% 3-year BCSS improved over time (64.9% to 80.7%; $p < 0.0001$) | NR |
| Chin et al., (2000) [18] | 44.0 (14–120) | Median survival: 22.0 mos (1–120 mos) | BRC: 52% (12 IBC-related deaths) NBRC: NR | 52% (12/23 patients) | Median DFS: 19.0 mos (1.0–120 mos) 16 ptx: 6 LRR, 10 distant |
| Hoffman et al., (2021) [19] | IBRC 42.9 (24.4–76.3) NBRC 45.4 (23.7–80.9) | 5-year OS 64.3% BRC vs. 57.2% NBRC IBRC vs. NBRC Adjusted OS (HR 0.63, 95% CI 0.44–0.90, $p = 0.01$) PM cohort OS (0.60, 95% CI 0.40–0.92, $p = 0.02$) | NR | NR | NR |
| Karadsheh et al., (2021) [13] | NR | Median unadjusted OS BRC—93.7 mos (95% CI 72.2–117.5) NBRC—68.1 mos (95% CI 65.5–71.7) $p < 0.001$ | NR | NR | NR |
| Nair et al., (2022) [23] | Mean, 68.4 (±46.8) | NR | NR | Crude (10-year survival): IBRC (62.9%), NBRC (47.6%) PM analysis (10-year survival): BRC (56.6%) and NBRC (62.2%), not significant | NR |
| Nakhlis et al., (2019) [20] | 66.0 | Median survival: 87 mos (<1–212 mos) | NR | NR | Median DFS: 35 mos (<1–212 mos) 105 NBRC ptx after median follow-up: 66 mos (<1–212 mos) −6 LRR only −44 LRR + distant −55 Distant only 22/40 BRC ptx (12 IBRC, 10 DBRC) with median follow-up: 78 mos (7–191 mos) −3 LRR only −5 LRR + distant −14 distant |

**Table 6.** *Cont.*

| Study | Median Follow-Up (Months) | OS | OM | BCSS | Recurrence |
|---|---|---|---|---|---|
| Patel et al., (2018) [21] | NR | NR | Cumulative incidence of OM lower amongst IR patients (*p* = 0.013) | No difference between IBR status in BCSM (sHR 1.04; CI 0.71–1.54; *p* = 0.83) or adjusted BCSM (sHR 1.13; CI 0.84–1.93; *p* = 0.058) IBR did not influence cumulative incidence of BCSM | NR |
| Simpson et al., (2016) [22] | 2.3 years (1.4–4.6 yrs) | 1-year OS 94.7% (95% CI 30.0–56.9) 2-year OS 76.5% (95% CI 65.6–89.2) | 22 ptx NBRC: 14 IBRC: 7 DBRC: 1 Median time to death 21.9 mos | NR | Median DFS: 9.9 mos 26 ptx overall NBRC: 18 ptx —LRR: 4 —Distant: 14 IBRC: 7 ptx —LRR: 0 —Distant: 7 DBRC: 1 ptx —LRR: 0 —distant: 1 Recurrence rate 1-year 30.9% (95% CI 17.9–48.1) 2-year 45.1% (95% CI 30.0–56.9) |

Abbreviations: BCSS: breast cancer-specific survival; BCSM: breast cancer-specific mortality; BRC: breast reconstruction; CI: confidence interval; DBRC: delayed breast reconstruction; DFS: disease-free survival; HR: hazard ratio; IBC: inflammatory breast cancer; IBRC: immediate breast reconstruction; LRR: locoregional reccurence; mos: months; NBRC: no breast reconstruction; NR: not reported; OM: overall mortality; OS: overall survival; PM: propensity-matched; ptx: patients; SBRC: single breast reconstruction; SMX: single ipsilateral mastectomy; XRT: radiation therapy.

Chang et al. found breast reconstruction was associated with better survival (*p* = 0.004) [17]. In the study by Hoffman et al., inverse probability of treatment weighting analysis found no association, but multivariate analysis and propensity score matching showed breast reconstruction was associated with improved OS (HR 0.63, 95% CI 0.44–0.90, *p* = 0.01) [19]. The study by Patel et al. found cumulative incidence of OM to be lower amongst patients with breast reconstruction (*p* = 0.013) [21].

Six studies reported on predictors of OS (Table 7) [13,14,18,20,22,23] but in general, there is a lack of data in this regard. Chen et al. reported younger age, increasing year of diagnosis, non-black race, and limited comorbidities as positive predictors of OS [14], while Chin et al. reported a positive surgical margin increased the risk of mortality [18]. Chen et al. also reported that lower histologic grade, lower nodal stage, positive hormonal status, and greater than 10 lymph node removals were associated with better OS [14]. Chen et al. also found no significant difference between women who underwent a single mastectomy with breast reconstruction and those who chose to have a contralateral prophylactic mastectomy and breast reconstruction [14] (Table 7).

**Table 7.** Factors predicting improved oncologic outcomes.

| Study | OS | OM | BCSS | Recurrence |
|---|---|---|---|---|
| Chang et al., (2015) [17] | BRC improved OS compared to NBRC (*p* = 0.004) | NR | NR | NR |
| Chen et al., (2017) [14] | In UV and MV analysis: younger age, increasing year of diagnosis, non-black race, married status, low histologic grade, lower N stage, positive hormonal status, 10+ lymph node removal | NR | In UV and MV analysis: younger age, increasing year of diagnosis, non-black race, married status, low histologic grade, lower N stage, positive hormonal status, 10+ lymph node removal | NR |
| Chin et al., (2000) [18] | Positive surgical margin decreased OS (*p* = 0.02) | NR | NR | Positive surgical margin increased local recurrence (*p* = 0.02) |
| Karadsheh et al., (2021) [13] | BRC vs. NBRC Unadjusted HR 0.79 (95% CI 0.72–0.88, *p* < 0.001) Adjusted HR 0.95 (95% CI 0.85–1.06; *p* = 0.35) | NR | NR | NR |
| Nair et al., (2022) [23] | NR | NR | Crude: BCSS higher in NBRC (HR 0.72, 95% CI 0.60–0.86; *p* < 0.001) PM analysis: No difference in BCSS HR 0.96, 95% CI 0.79–1.16; *p* = 0.65 | NR |
| Patel et al., (2018) [21] | NR | No difference in OM by race, US region, poverty, median income, year of IBC diagnosis. UV and MV analysis: BRC not associated with lower OM (HR = 0.82, CI 0.55–1.21; *p* = 0.319). Independent predictors of worse OM: older age, higher CCI, single or widowed, negative or unknown hormone receptor status, no or unknown # LN examined, and increasing # positive LN (*p* < 0.0001) Poor histologic grade (*p* = 0.0318) No radiation received (*p* = 0.0066), No chemotherapy received (*p* = 0.0343). | BCSS not associated with age of diagnosis, race, marital status, US region, SES factors, CCI, or XRT UV and MV analysis: BRC not associated with increased BCSS (HR = 1.14, CI 0.71–1.76; *p* = 0.55). Independent predictors of worse BCSS: Earlier diagnosis year (*p* = 0.0003) Poor or intermediate histologic grade (*p* = 0.0005) ER/PR negative or unknown, increasing # positive LN (*p* < 0.0001) Receiving CTX (*p* = 0.0006). | NR |
| Simpson et al., (2016) [22] | NR | BRC not associated with increased mortality *p* = 0.91 | NR | NR |

Abbreviations: #: number; BCSS: breast cancer-specific survival; BRC: breast reconstruction; CI: confidence interval; CCI: Charlson comorbidity index; CPM: contralateral prophylactic mastectomy; CTX: chemotherapy; HR: hazard ratio; IBC: inflammatory breast cancer; IBRC: immediate breast reconstruction; IPW: inverse-probability weighting; LN: lymph node; mos: months; MV: multivariate; NBRC: no breast reconstruction; OM: overall mortality; OS: overall survival; PM: propensity-matched; SES: socioeconomic; SMX: single mastectomy; UV: univariate. Hoffman et al. (2021) [19], Karadsheh et al. (2021) [13], and Nakhlis et al. (2019) [20] did not report on predictors of oncologic outcomes.

### 3.8. Breast-Cancer Specific Survival (BCSS)

Three studies reported the impact of breast reconstruction on BCSS, finding no statistically significant association [14,21,23] (Table 6). Chen et al. [14] found 3-year BCSS to improve over time (64.9% to 80.7%; *p* < 0.0001) with a 5-year BCSS rate of 59%. There was also no difference between patients who chose to undergo a single mastectomy with breast reconstruction or double mastectomy (single mastectomy with contralateral prophylactic mastectomy with breast reconstruction) compared to a single unilateral mastectomy without reconstruction alone. In multivariate analysis, sociodemographic (younger age, increasing diagnostic year, race, marital status) and tumor/pathology-specific factors (lower histologic grade, lower nodal status, positive hormonal status, and removal of 10+ lymph

nodes) were predictors for better BCSS [14]. Patel et al. found no association between breast reconstruction and breast-cancer specific mortality (*p* = 0.058) [21]. Demographic and socioeconomic factors, except for the increasing diagnostic year, were not associated with BCSS. Tumor and pathologic factors, however, were associated with BCSS, with poor or intermediate histologic grade, unknown or negative ER/PR status, increasing number of positive lymph nodes and receiving chemotherapy predicting worse survival (Table 7). We conducted a propensity-matched analysis between reconstruction and no-reconstruction patients and found no difference in BCSS (HR 0.96, 95% CI 0.79–1.16; *p* = 0.65) [23].

*3.9. Recurrence*

Four studies commented on disease recurrence following breast reconstruction [17,18,20,22] (Table 6). The median disease-free survival ranged from 9.9–35 months for the overall cohort [17,18,20,22]. Simpson et al. reported 1-year and 2-year recurrence rates of 30.9% (95% CI 17.9–48.1) and 45.1% (95% CI, 30.0–56.9), respectively. The patients predominantly belonged to the non-reconstructive group and experienced distant disease metastasis [22]. The trend was similar among the other included studies. Based on the limited data from Simpson et al. [22] and Nakhlis et al. [20], there appears to be no difference in the risk of recurrence (locoregional or distant) between non-reconstructive and reconstructive groups. Rates of recurrence were also similar between patients having immediate versus delayed breast reconstruction, with the majority having a distant recurrence in their studies. The only predictor for increased risk of local recurrence was found to be positive surgical margins by Chin et al., (*p* = 0.02) [18].

## 4. Discussion

IBC patients are generally discouraged from receiving immediate breast reconstruction after mastectomy due to concerns with long-term survival, delays in initiating adjuvant radiation therapy, and disease recurrence. There have also been concerns that a reconstructed breast mound might render delivering adjuvant radiation more challenging [6,7]. However, with advances in systemic therapy, the survival of women with IBC continues to improve, with recent OS rates reported between 55–71% at 2–5 years [24–27].

The population-based studies reported a significant increase in the use of immediate breast reconstruction among IBC patients over time, which parallel an increase in CPM. CPM is also discouraged at the time of initial surgery in IBC due to the increased risks of developing surgical complications that may also delay the initiation of adjuvant treatment and lack of survival benefit [6,7,15]. Despite this increase in use, our review found no significant difference in OS or BCSS between women with IBC who had and did not have breast reconstruction. The population-based studies only looked at immediate reconstruction, which remains a relative contraindication in the setting of IBC [15]. We surmise that there was no impact of immediate breast reconstruction on survival because women with IBC typically succumb to distant disease and locoregional management may not impact the development of distant disease. Two studies found breast reconstruction to be associated with improved OS [17,19]; however, the study by Chang et al. [17] included only delayed breast reconstruction, which has an inherent selection bias. The second study by Hoffman et al. [19] looked at immediate breast reconstruction and found better overall survival, even after propensity-matched analysis. This is an interesting finding that is hypothesis-generating, but given the retrospective data, there may be other confounding factors and selection biases that were not accounted for.

As with non-IBC patients, we found that demographic and socioeconomic factors highly predict receipt of breast reconstruction in IBC patients [28–31]. Younger age at diagnosis is a well-known predictor [32–34]. Receiving a CPM also predicts breast reconstruction because the two procedures often occur in tandem [14,19,23], despite the lack of survival benefit for CPM in women with unilateral, nonhereditary breast cancer [35–38]. Lack of private insurance, low income and education, and rural residence appear to be barriers to accessing breast reconstruction [28–31]. A better effort is needed to promote access and use of breast reconstruction amongst socioeconomically disadvantaged women.

Race or ethnicity did not influence the use of breast reconstruction amongst IBC patients at the population level, but we surmise that there is likely limited data in the IBC literature evaluating breast reconstruction amongst minority communities. Clinicopathologic factors were not uniformly found to predict reconstruction use in IBC patients across studies.

Complication rates following immediate breast reconstruction in IBC patients appear comparable with those in non-IBC patients [39–41]. Notably, Hoffman et al. found no difference in 30- and 90-day mortality and 30-day readmission rates between immediate reconstruction and no reconstruction [19], suggesting that for select patients, mastectomy with immediate reconstruction is a feasible procedure with minimal impact on short-term surgical morbidity and mortality [42,43]. There was no clinically significant difference in time to post-mastectomy radiotherapy (PMRT) between the breast reconstruction and no-reconstruction groups, although the data is limited. Regardless of reconstruction status, women still initiated their PMRT within the recommended optimal timeline of 12 weeks [44–46], similar to the findings reported in non-IBC patients [47,48].

Rates of recurrence in our study are consistent with the literature on IBC, with 3- to 5-year cumulative rates ranging between 17–65% [5,24,26,49]. In our study, distant metastases were consistently higher than LRR, with recurrence rates of 38.3% and 56.1% in the reconstruction and non-reconstruction groups, respectively. The high distant recurrence rates speak to the aggressive nature of IBC [5]. Others have reported potential concerns about maintaining some of the skin envelope that is often used for immediate alloplastic reconstruction as a deterrent from immediate breast reconstruction [15,49,50]. In review of the limited data, local recurrence rates are similar between immediate and delayed reconstruction, but actual sample sizes are very small to reach any definitive conclusion.

The strength of our systematic review lies in it being the first review, to our knowledge, to evaluate the literature on outcomes after breast reconstruction in women with IBC. As the survival of women with IBC continues to improve, we anticipate that more women with IBC will consider breast reconstruction. We identified a limited number of studies, all retrospective, at the institutional and population-based level on this topic. The studies vary in the extent to which immediate versus delayed reconstruction patients are represented. Most of the large population-based databases using SEER and NCDB report immediate breast reconstruction outcomes, and therefore our conclusions are restricted largely to IBC patients having immediate breast reconstruction. There is no prospective data on the safety of immediate breast reconstruction in IBC [50]. At this time, delayed post-mastectomy reconstruction with an autologous flap remains the recommended reconstruction for IBC patients [15]. The population-level data suggests that immediate breast reconstruction may not impact oncologic outcomes; however, we acknowledge the possibility that patients with a more favorable disease burden or biology were more likely to undergo breast reconstruction than those with a poor disease burden or biology. We highlight that there is no prospective data, and there are limitations to interpreting retrospective data.

There are limitations to our study, including the retrospective study design of the included studies and its inherent biases. There are inconsistencies in the data reported across studies due to variations in the data collection process, as limited by the prolonged follow-up time, patient databases, or data sources. For example, while some studies distinguished between immediate versus delayed reconstruction or the treatment status of patients before and after mastectomy, other studies were less clear on this data. Consequently, the heterogenous reporting precluded the ability to conduct a meta-analysis. Prospective studies using objective selection criteria regarding patient and tumor characteristics minimize potential sources of bias and are needed in this area. Secondly, all studies included in the study were either conducted in the United States or used patient information from cancer databases based in the United States. The paucity of literature from outside the United States and low-income countries limits the generalizability of the results. Lastly, given that the patient information is sourced from similar national databases, the literature may be reporting on similar patients across multiple studies. The nuanced variations in data collection and analyses between the studies offered unique findings on the topic, warranting the need to

include these studies. An important limitation that has been previously acknowledged by others is that it is unclear if all patients diagnosed as IBC truly had the disease, since the diagnosis of IBC is clinical and some of these patients may in fact have had non-IBC locally advanced breast cancer [15]. Data on the specific type of reconstruction (autologous versus alloplastic) is incomplete in SEER. There is a lack of data on delayed reconstruction in IBC since the population-based registries only capture immediate reconstruction.

## 5. Conclusions

Despite the lack of safety data, there is an increasing use of immediate breast reconstruction among women with IBC. Predictors of immediate breast reconstruction are largely sociodemographic and include younger age, having private insurance, higher median income, and education level, and living in a metropolitan setting. The limited available retrospective data does not indicate a worse oncologic outcome between women who undergo and do not undergo immediate breast reconstruction. Immediate breast reconstruction may be an appropriate consideration for some IBC patients who desire the procedure; however, there remains a paucity of prospective data on the safety of immediate breast reconstruction in this setting. Future research should focus on characterizing the safety of immediate breast reconstruction through prospective multicenter studies, as well as the selecting the appropriate patient candidates.

**Supplementary Materials:** The following supporting information can be downloaded at: https://www.mdpi.com/article/10.3390/curroncol30070489/s1, Table S1: Search Strategy.

**Author Contributions:** A.G.N. and D.W.L. contributed to the study design and conception. A.G.N. and G.T.Y.K. performed the literature search. A.G.N., G.T.Y.K., J.L.S and D.W.L. performed data analysis and interpretation. A.G.N. drafted the manuscript with review by G.T.Y.K., J.L.S. and D.W.L. All authors have read and agreed to the published version of the manuscript.

**Funding:** This research received no external funding.

**Institutional Review Board Statement:** Ethical review and approval were waived for this study, due to this study being a review of published/publicly reported literature.

**Informed Consent Statement:** Patient consent was waived due to this study being a review of published/publicly reported literature.

**Data Availability Statement:** The data presented in this study are outlined in our search strategy and reference list.

**Acknowledgments:** D. Lim is supported by the Canadian Cancer Society Chair in Breast Cancer Research at Women's College Research Institute of Women's College Hospital (Toronto, ON, Canada).

**Conflicts of Interest:** A.G. Nair, G.T.Y. Ko and J.L. Semple have no conflict of interest. D.W. Lim has received consulting fees from Astra Zeneca and Merck & Co., Inc. for systemic therapies unrelated to this study. The authors declare that they have no known competing financial interests or personal relationships that could have appeared to influence the work reported in this paper.

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
