# Peer review of "Breast Reconstruction Use and Impact on Surgical and Oncologic Outcomes Amongst Inflammatory Breast Cancer Patients—A Systematic Reviewâ€"

_curroncol, doi:10.3390/curroncol30070489_

Round 1
Reviewer 1 Report (New Reviewer)
Thank you for the resubmission of your revised paper. The changes made have improved the manuscript.
Please add a definition for Medicare and Medicaid in Table 1 as the difference between American public insurance policies will not be known to most readers outside of the USA.
Author Response
Reviewer: 1
Comments and Suggestions for Authors
Thank you for the resubmission of your revised paper. The changes made have improved the manuscript. Please add a definition for Medicare and Medicaid in Table 1 as the difference between American public insurance policies will not be known to most readers outside of the USA.
- Thank you for the suggestion. We have added the definition to the table legend (highlighted, lines 140-141)
Reviewer 2 Report (New Reviewer)
This is a systematic review of the use of Immediate Breast Reconstruction following Mastectomy for Inflammatory Breast Cancer ( IBC). It focuses on the oncologic outcomes and also addresses surgical complications in terms of them being the source of potential treatment delay.
Methodology.
The analysis/comparison tool is the Newcastle Ottawa quality assessment tool. The selection of studies seems appropriate.
Results
- there is largely a "no difference detected" in OS, DSS and treatment delays between all of the groups, with all of the caveats around Type II errors
There was a surprising survival benefit for immediate reconstruction in some studies - this would be hypothesis generating
Discussion
- would the authors like to postulate why there was no difference in survival between the cohorts - is there truly no negative effect of surgery at that timing od the treatemnt?
- is it possible that the patients selected for Immediate reconstruction in all of the studies had more favourable disease? ( more pCR?), hiding a potential harm of immed reconstruction?
- can the authors confirm the recommended (or routinely recommended) use of adjuvant external beam Radiotherapy after mastectomy in these patients?
- if this is the case, was there any ability to quantify the cosmetic outcomes ? in many centres, post mastectomy radiotherapy is a contra-indication for immediate reconstruction, whereas "reverse sequenced" pre mastectomy is not
Tables and figures
appropriate
Author Response
Reviewer: 2
Comments and Suggestions for Authors
This is a systematic review of the use of Immediate Breast Reconstruction following Mastectomy for Inflammatory Breast Cancer (IBC). It focuses on the oncologic outcomes and addresses surgical complications in terms of them being the source of potential treatment delay.
Methodology: The analysis/comparison tool is the Newcastle Ottawa quality assessment tool. The selection of studies seems appropriate.
- Thank you.
Results
There is largely a "no difference detected" in OS, DSS and treatment delays between all of the groups, with all of the caveats around Type II errors
There was a surprising survival benefit for immediate reconstruction in some studies - this would be hypothesis generating
- We agree that this would be hypothesis-generating and may relate to selection bias in the type of women being offered breast reconstruction. In our study using SEER data, we tried to eliminate the effect of potential selection bias by performing a propensity-matched analysis and found no difference in breast cancer-specific survival between women who had immediate reconstruction and those who did not (reference 23).
Discussion
Would the authors like to postulate why there was no difference in survival between the cohorts - is there truly no negative effect of surgery at that timing of the treatment?
- We surmise that there is no difference in survival because inflammatory breast cancer patients typically succumb to distant disease, and so what we do locoregionally would not change that. However, women with IBC are living longer because of advances in systemic therapy of IBC (lines 327-328), which may justify those who seek breast reconstruction to augment their quality of life. We have added this hypothesis to our discussion [highlighted, lines 335-338].
Is it possible that the patients selected for Immediate reconstruction in all the studies had more favourable disease? (more pCR?), hiding a potential harm of immediate reconstruction?
- For population-based studies that used the SEER database (like our reference #23), pCR rates are not available in SEER. In contrast, the NCDB reports on pathologic stage. For the two studies that used NCDB, Hoffmann et al. (2021) did not find that pathologic stage predicted immediate reconstruction rates while Karadsheh et al. (2021) found that lower pathologic stage and nodal status predicted immediate breast reconstruction. Despite this discrepancy, both studies reported better overall survival with immediate breast reconstruction compared to no breast reconstruction. Hoffmann et al. also performed propensity-matched analysis and still found better overall survival with breast reconstruction. This suggests that patients selected for immediate reconstruction possibly had more favorable disease (more response to neoadjuvant chemotherapy), but breast reconstruction did not cause detriment to their overall survival compared with a similarly matched cohort of patients who didn’t have breast reconstruction. We have now add in Table 4 that Karadsheh et al. specifically found lower pathologic stage and lower pathologic nodal stage to be associated with breast reconstruction (highlighted).
Can the authors confirm the recommended (or routinely recommended) use of adjuvant external beam radiotherapy after mastectomy in these patients?
- The studies included did include patients who received radiation. For example, Karadsheh et al. documented the number of patients who received chemotherapy and radiation. Hoffmann et al. also include the time to PMRT as a variable.
If this is the case, was there any ability to quantify the cosmetic outcomes? in many centres, post mastectomy radiotherapy is a contra-indication for immediate reconstruction, whereas "reverse sequenced" pre mastectomy is not
- Cosmetic outcomes unfortunately were not reported in the studies included in this systematic review, as it is not collected in SEER and NCDB. In general, there is a lack of data regarding cosmetic outcomes following inflammatory breast cancer patients undergoing reconstruction. We would consider PMRT a relative contraindication for immediate reconstruction and certainly in our centers (University Health Network & Women’s College Hospital, Toronto, Ontario, Canada), we routinely perform autologous (DIEP) and alloplastic (1 stage or 2 stage with a tissue expander) in patients who are expected to receive PMRT.
Tables and figures appropriate
- Thank you
Round 2
Reviewer 1 Report (New Reviewer)
Thank you for adding the definitions of Medicare and Medicaid as requested.
Author Response
Thank you for your comments.
This manuscript is a resubmission of an earlier submission. The following is a list of the peer review reports and author responses from that submission.
Round 1
Reviewer 1 Report
Thank you for providing an extensive systematic review on Breast Reconstruction after IBC. I applaud the amount of effort that has gone into researching this topic. Although interesting, there are several flaws in the presented manuscript. It aims to describe trends, predictors and survival of women undergoing BR, yet the included studies have varying follow-up, and the risk of bias assessment is not utilized during the analysis of results. Also, it is unclear if the actual results describe immediate or delayed or a combination of both (although the introduction explicitly states immediate BR).
Methods: Great to see a quality assessment of the papers were done via the risk for bias; yet this has not been utilized during the results
were the PRISMA guidelines utilized? If so, please state this.
Please consider registering your systematic reviews beforehand via PROSPERO
A second reviewer would improve the quality assurance of the manuscript during TIAB screening.
Results:
Figure 1 is missing.
Table 2 is inconsistent wrt the legend and abbreviations
Table 4 is unclear and too long
Write the text and use the tables as evidence to support your results
Type of breast reconstruction not well reported on
Discussion: new information is being presented; keep this in the results.
Conclusion: the aim(s) of the study was incongruent with the conclusion should be reconsidered. Only survival rate is mentioned (yet flawed due to the articles with have only a very short follow-up), no trends nor predictors are mentioned
Future research should focus on identifying optimal candidates --> the abstract reads this as something that has already been performed
Reviewer 2 Report
Dear authors,
Your manuscript is well-written, but you have too many self citations. I understand that is a review, but in the same time you cited a lot from your previous work.
Reviewer 3 Report
Dear Authors,
I reviewed the study titled "Breast Reconstruction Use and Impact on Surgical and Oncologic Outcomes Amongst Inflammatory Breast Cancer Patients - A Systematic Review". Thank you for preparing a comprehensive study on the Use of Breast Reconstruction and Its Impact on Surgical and Oncologic Outcomes in Inflammatory Breast Cancer Patients. Your review has been covered quite adequately, Good work.
Yours sincerely